# Clinical course and potential predictive factors for pneumonia of adult patients with Coronavirus Disease 2019 (COVID-19): A retrospective observational analysis of 193 confirmed cases in Thailand

**Wannarat A. Pongpirul**[1☯], **Surasak Wiboonchutikul**[1☯]*, **Lantharita Charoenpong**[1], **Nayot Panitantum**[1], **Apichart Vachiraphan**[1], **Sumonmal Uttayamakul**[1], **Krit Pongpirul**[2,3], **Weerawat Manosuthi**[1‡], **Wisit Prasithsirikul**[1‡]

1 Bamrasnaradura Infectious Diseases Institute, Department of Disease Control, Ministry of Public Health, Nonthaburi, Thailand, 2 Department of Preventive and Social Medicine, Faculty of Medicine, Chulalongkorn University, Bangkok, Thailand, 3 Department of International Health, Johns Hopkins Bloomberg School of Public Health, Baltimore, Maryland, United States of America

☯ These authors contributed equally to this work.
‡ These authors are joint senior authors on this work.
* surasakwiboon@gmail.com

## Abstract

Clinical spectrum of Coronavirus Disease 2019 (COVID-19) remains unclear, especially with regard to the presence of pneumonia. We aimed to describe the clinical course and final outcomes of adult patients with laboratory-confirmed COVID-19 in the full spectrum of disease severity. We also aimed to identify potential predictive factors for COVID-19 pneumonia. We conducted a retrospective study among adult patients with laboratory-confirmed COVID-19 who were hospitalized at Bamrasnaradura Infectious Diseases Institute, Thailand, between January 8 and April 16, 2020. One-hundred-and-ninety-three patients were included. The median (IQR) age was 37.0 (29.0–53.0) years, and 58.5% were male. The median (IQR) incubation period was 5.5 (3.0–8.0) days. More than half (56%) of the patients were mild disease severity, 22% were moderate, 14% were severe, and 3% were critical. Asymptomatic infection was found in 5%. The final clinical outcomes in 189 (97.9%) were recovered and 4 (2.1%) were deceased. The incidence of pneumonia was 39%. The median (IQR) time from onset of illness to pneumonia detection was 7.0 (5.0–9.0) days. Bilateral pneumonia was more prevalent than unilateral pneumonia. In multivariable logistic regression, increasing age (OR 2.55 per 10-year increase from 30 years old; 95% CI, 1.67–3.90; p<0.001), obesity (OR 8.74; 95%CI, 2.06–37.18; p = 0.003), and higher temperature at presentation (OR 4.59 per 1°C increase from 37.2°C; 95% CI, 2.30–9.17; p<0.001) were potential predictive factors for COVID-19 pneumonia. Across the spectrum of disease severities, most patients with COVID-19 in our cohort had good final clinical outcomes. COVID-19 pneumonia was found in one-third of them. Older age, obesity, and higher fever at presentation were independent predictors of COVID-19 pneumonia.

**Data Availability Statement:** All relevant data are within the manuscript and its Supporting Information files.

**Funding:** The authors received no specific funding for this work.

**Competing interests:** The authors have declared that no competing interests exist.

## Author summary

This report describes the clinical course and final clinical outcomes of the full spectrum of disease severity of Coronavirus Disease 2019 (COVID-19) and evaluates risk factors of pneumonia in 193 laboratory-confirmed cases of COVID-19 in the Bamrasnaradura Infectious Diseases Institute, Thailand. The majority of patients with COVID-19 had mild disease. COVID-19 pneumonia was found in approximately 40% of patients. Older age, obesity, and higher grade of fever at presentation were independent risk factors of pneumonia in adult patients with COVID-19. Most patients in our cohort recovered and were discharged from hospital (recovery rate, 98%). Our findings can help the public health systems to estimate the burden of the disease and identify vulnerable patients in a timely manner.

## Introduction

Coronavirus Disease 2019 (COVID-19) is the most recent emerging infectious disease, and it is caused by novel Severe Acute Respiratory Syndrome Coronavirus-2 (SARS-CoV-2) [1]. COVID-19 was first identified in China in December 2019 [2], and has become a global menace with a great impact on the health systems of affected countries. Several studies have described the demographic and clinical characteristics, disease severity, and treatment outcomes of patients with COVID-19 [3–8]. These reports focused on the findings of patients with moderate and severe diseases and most of the disease severity assessments were based on evaluation at the time of admission, which were likely for triage purpose. The temporal clinical progression during hospitalization was well documented [9, 10], disease severity at admission can change during hospital stay and can also differ from that at discharge. Also, the final clinical outcomes defined as recovered or deceased in the published reports could not be fully established due to the fact that a number of the patients were still hospitalized at the time of data analysis [3–6, 11, 12]. Thus, a complete picture and the final clinical outcome, especially the rate of recovery, of patients with COVID-19 are still uncertain.

COVID-19 has been categorized into a range of clinical severity including, asymptomatic, mild, moderate (non-severe pneumonia), severe (severe pneumonia), and critical illness [13, 14]. Data from a large cohort from China showed 81% of patients had a mild disease while 14% were severe and 5% developed critical illness [15]. Nevertheless, patients defined as 'mild' in the Chinese nationwide survey varied from having minimal symptoms without lung involvement to having early pneumonia. Data on COVID-19 patients with both mild disease and mild pneumonia are lacking. Cases defined as 'non-severe' were excluded in a recent randomized controlled trial on favipiravir [16]. The incidence and risk factors of pneumonia of any severity in SARS-CoV-2-infected patients are also unknown. Understanding the full spectrum of COVID-19, rather than only the more severe end of the disease, would facilitate public health systems to estimate the burden of the disease and to identify vulnerable patients earlier.

On January 13, 2020, Thailand reported a confirmed case of COVID-19, the first recorded case outside of China [17]. This case was admitted to the Bamrasnaradura Infectious Diseases (BIDI), which the Thai Ministry of Public Health designated as the national infectious disease referral hospital for the COVID-19 outbreak. As of May 29, 2020, 3,076 patients with confirmed cases of COVID-19 had been reported in Thailand [18].

This study aimed to present details of all adult hospitalized patients with laboratory-confirmed COVID-19 who were admitted to our institute regardless of the severity of their

disease. We described the clinical course and final outcome (recovered or deceased) of the disease. Potential predictive factors of COVID-19 pneumonia were also investigated.

## Methods

### Ethics statement

The study was reviewed and approved by the BIDI's Institutional Review Board (S012h_63_ExPD). Informed consent was waived due to de-identification of patient data.

### Patients

BIDI is the main public health institution under the Department of Disease Control, Ministry of Public Health of Thailand responsible for testing and treating emerging infectious diseases including COVID-19. All individuals who were diagnosed as COVID-19, according to the WHO interim guidance [13] were admitted at the institute, regardless of the severity of their disease. BIDI's protocol requires nasopharyngeal and throat swab samples to be obtained to test for SARS-CoV-2 by real-time reverse-transcription–polymerase-chain-reaction (RT-PCR) assay at two-day intervals during hospitalization, until two consecutive negative results at least 24 hours apart were achieved. Baseline chest radiograph was performed in every patient at admission. The need for follow-up chest radiograph during admission was based on the judgement of each attending physicians. The patients were discharged if they met the following criteria: 1) resolution of fever without the use of antipyretics $\geq$ 48 hours, 2) improvement in respiratory symptoms with oxygen saturation $\geq$ 95% while they were breathing ambient air, and 3) samples from nasopharyngeal and throat swab tested negative for SARS-CoV-2 by real-time RT-PCR. We conducted a retrospective cohort study among all adult patients aged $\geq$ 18 years with laboratory-confirmed COVID-19 who were hospitalized at BIDI, between January 8 and April 16, 2020. The hospital outcomes were monitored until discharges or death.

### Definitions

A laboratory-confirmed COVID-19 was defined as detecting SARS-CoV-2 RNA in nasopharyngeal and throat swab specimens by RT-PCR assay. Fever was defined as an axillary temperature of $\geq$ 37.3˚C. Defervescence was defined as resolution of fever (axillary temperature < 37.3˚C) without the use of fever-reducing medications. Pneumonia was diagnosed by the presence of respiratory symptoms and opacity on chest radiography. Pneumonia with detection of SARS-CoV-2 RNA from respiratory specimens was considered as COVID 19-associated pneumonia. Acute respiratory distress syndrome (ARDS) was determined according to the Berlin definition [19]. Acute kidney injury (AKI) was defined according to Kidney Disease Improving Global Outcomes (KDIGO) guideline [20]. Obesity was classified as body mass index (BMI) $\geq$ 30 kg/m$^2$ according to World Health Organization (WHO) classification for overweight and obesity [21]

### Real-time reverse transcription-polymerase chain reaction assay for SARS-CoV-2

Respiratory specimens were collected from the nasopharynx and oropharynx using synthetic fiber or flocked swabs. The swabs from both sites were placed in the same tube to increase viral detection. Samples were transported in a viral transport medium containing anti-fungal and antibiotic supplements were used. Sputum specimens were collected from patients with lower respiratory symptoms.

Total nucleic acid or viral RNA was extracted from the specimens and tested with conventional nested RT-PCR for coronavirus family of the first two novel coronavirus cases in Thailand. Both cases were confirmed as Wuhan human novel coronavirus 2019 by two reference laboratories—the Thailand National Institute of Health, Ministry of Public Health and Emerging Infectious Disease Health Sciences Center, King Chulalongkorn Memorial Hospital, Thai Red Cross Society—using whole-genome sequencing comparison to the Wuhan reference virus (posted in GenBank, accession number MN908947). After Wuhan human novel coronavirus 2019 sequence data were available, real-time RT-PCR with SARS-CoV-2-specific primers and probes were developed to detect the specific gene areas using the WHO protocol. Real-time RT-PCR testing was based on fluorescent PCR and probes consist of a reporter dye and quencher dye. The PCR instrument was automatically amplified and detect the fluorescent signal. To avoid contamination, non-template or negative controls were included in every PCR run. Human housekeeping gene was used as an internal control to monitor the process of specimen collection and extraction. To confirm COVID-19 infection at the early phase of the COVID-19 outbreak in January and February 2020, SARS-CoV-2 RNA had to be detected by two independent laboratories.

At BIDI, we used two real-time RT-PCR techniques to detect SARS-CoV-2. First, the COVID-19 Coronavirus Real Time PCR Kit (Jiangsu Bioperfectus Technologies Co.,Ltd.; WHO Product Code JC10223-1NW-25T or JC10223-1NW-25T) was used for detecting the Open Reading Frame gene region (ORF 1ab) and viral nucleocapsid region (N gene) according to the recommendation of the Chinese CDC. Second, the Real-Time Fluorescence Detection RT-PCR kit (BGI technology) was concurrently used for detecting the ORF 1ab gene. After March 31, 2020, Cobas SARS-CoV-2 qualitative assay for use on the Cobas6800/8800 Systems (Roche Molecular Systems, Inc.) was used at BIDI. Both ORF 1ab and E gene were designed for SARS-CoV-2 detection according to WHO recommendation.

## Data collection

Demographic, epidemiological, clinical, hospital courses, investigation, and treatment data of all consecutive laboratory-confirmed cases were extracted from medical records reviewed by four attending physicians responsible for the patients with COVID-19 at BIDI. Chest radiography interpretation was based on reports by radiologists and independently rechecked by a pulmonologist during the data extraction. The severity of illness of each patient was evaluated at the time of discharge or death by the study team.

## Clinical outcomes

The patients who met the discharged criteria were defined as recovered. All laboratory-confirmed cases who died during hospitalization regardless of any negative follow-up RT-PCR results defined as deceased. The severity of illness of each patient was classified following the report of the WHO-China Joint mission on Coronavirus Disease 2019 [22]: mild (the clinical symptoms were mild, and there was no sign of pneumonia on imaging), moderate (fever and respiratory symptoms with radiological findings of pneumonia, but without features of severe pneumonia), severe (respiratory rate $\geq$30 breaths/minute, oxygen saturation $\leq$93%, PaO$_2$/FiO$_2$ ratio <300, and/or lung infiltrates >50% of the lung field within 24–48 hours), and critical (respiratory failure, shock, and/or multiple organ failure). Asymptomatic infection was defined as when patients had no symptoms or signs throughout the course of the disease. Patients were categorized into two groups based on pneumonia detection (pneumonia vs non-pneumonia).

## Statistical analysis

Descriptive data are presented as mean and standard deviation (SD), median and interquartile range (IQR), and frequencies (%), as appropriate. No imputation was made for missing data. The mean values of continuous variables with normal distribution between the pneumonia and non-pneumonia groups were compared using Student's *t-test*. Categorical variables between the pneumonia and non-pneumonia groups were compared using the Chi-squared test and Fisher's exact test, as appropriate. Logistic regression analysis was used to determine factors associated with pneumonia in patients with COVID-19. We excluded variables from the logistic model if their nature was highly subjective (presenting symptoms that were patient self-reported), if the data were not available $\geq$ 20% of all cases (blood chemistry results), or if they were correlated with pneumonia (e.g. high respiratory rate and low oxygen saturation). Variables with p-value < 0.05 on univariate analysis were included in the multiple logistic regression model. Collinearity diagnostics were performed for the multivariable logistic regression analysis. A correlation of > 0.5 was considered risk of bias estimation due to collinearity. All statistical analyses were performed using SPSS version 26.0 (IBM SPSS Statistics Subscription Trial). A p-value < 0.05 was considered statistically significant.

## Results

### Patients' characteristics

A total of 195 laboratory-confirmed SARS-CoV-2 infected patients were admitted to BIDI during the study period. This included 11 patients previously reported during the early phase of COVID-19 outbreak [23]. Of the 195 laboratory-confirmed SARS-CoV-2-infected patients, 193 (99.0%) had either a recovered or deceased final clinical outcome. The other two patients were referred to other hospitals before viral RNA clearance according to their requests to be referred. Two consecutive negative RT-PCR results from nasopharyngeal and throat swab were obtained in 82.4% and single negative results in 17.6%. The median (IQR) age of the patients was 37.0 (29.0–53.0) years, 58.5% were males, and 91.2% were Thai. The median BMI (IQR) was 23.3 (20.4–25.9) kg/m$^2$ and 12.7% were obese. One-quarter of the patients had one or more coexisting medical conditions, which was found less frequently in mild cases. Hypertension, diabetes, and dyslipidemia were the most common comorbidities. Of all cases, 79.3% were local transmission, and 20.7% were imported cases. The epidemiological data showed that 34.7% had a history of contact with a confirmed COVID-19 case, 20.7% had arrived from affected countries with widespread or ongoing transmission of COVID-19 within 14 days before the onset of illness, 17.1% had attended or worked at crowded places, 22.8% were involved with a boxing stadium cluster, and only one patient was linked with a healthcare facility (Table 1).

### At presentation

Contact investigation was able to identify the date of disease contact in 83 (43.0%) patients. Among this group, the median (IQR) incubation period among this group was 5.5 (3.0–8.0) days. The median (IQR) time from onset of illness to the first visit was 3.0 (2.0–6.0) days. Baseline clinical characteristics of the patients are shown in Table 1. Fever (62.7%) was the most common presenting symptoms, followed by dry cough (49.2%). Coryza, including rhinorrhea and sore throat, were reported in 28% of cases. Gastrointestinal symptoms were initially present in less than 10% of the patients. At presentation, only 78 (39.8%) of patients were found to have a fever by a thermometer measurement. The mean axillary temperature of mild cases was 37.0˚C, whereas those of moderate and severe cases were more than 37.5˚C. The patients with moderate and severe disease had laboratory abnormalities of greater magnitude (e.g. lower

**Table 1.  Baseline characteristics and initial findings of the study patients.**

| | All (n = 193) | Asymptomatic (n = 10) | Mild (n = 108) | Moderate (n = 43) | Severe (n = 26) | Critical (n = 6) |
|---|---|---|---|---|---|---|
| **Baseline characteristics** | | | | | | |
| Age, median (IQR), y | 37.0 (29.0–53.0) | 43.0 (31.3–56.3) | 32.0 (26.0–40.5) | 48.0 (34.0–59.0) | 52.5 (46.5–56.3) | 64.0 (41.8–72.3) |
| Age distribution, n (%) | | | | | | |
| 20–29 y | 53 (27.5) | 2 (20.0) | 44 (40.7) | 6 (14.0) | 1 (3.8) | 0 |
| 30–39 y | 53 (27.5) | 2 (20.0) | 37 (34.3) | 10 (23.3) | 3 (11.1) | 1 (16.7) |
| 40–49 y | 30 (15.5) | 3 (30.0) | 14 (13.3) | 8 (18.6) | 4 (15.4) | 1 (16.7) |
| 50–59 y | 34 (17.6) | 1 (10.0) | 10 (9.3) | 9 (20.9) | 13 (50.0) | 1 (16.7) |
| 60–69 y | 15 (7.8) | 2 (20.0) | 3 (2.8) | 9 (20.9) | 1 (3.8) | 0 |
| 70–79 y | 8 (4.1) | 0 | 0 | 1 (2.3) | 4 (15.4) | 3 (50.0) |
| Gender, n (%) | | | | | | |
| - Male | 113 (58.5) | 5 (50.0) | 53 (49.1) | 32 (74.4) | 17 (65.4) | 6 (100) |
| - Female | 80 (41.5) | 5 (50.0) | 55 (50.9) | 11 (25.6) | 9 (34.6) | 0 |
| BMI, median (IQR), kg/m$^2$ | 23.3 (20.4–25.9) | 22.8 (21.7–29.4) | 21.6 (19.3–24.6) | 25.4 (22.6–31.1) | 25.1 (22.9–29.8) | 24.2 (21.5–25.5) |
| Distribution of BMI (n = 173), n (%) | | | | | | |
| - <18.5 kg/m$^2$ | 17 (9.8) | 1 (14.3) | 15 (14.9) | 0 | 1 (4.2) | 0 |
| - 18.5–24.9 kg/m$^2$ | 99 (57.2) | 3 (42.9) | 66 (65.3) | 16 (45.7) | 10 (41.7) | 4 (66.7) |
| - 25.0–29.9 kg/m$^2$ | 35 (20.2) | 2 (28.6) | 15 (14.9) | 9 (25.7) | 7 (29.2) | 2 (33.3) |
| - $\geq$ 30.0 kg/m$^2$ | 22 (12.7) | 1 (14.3) | 5 (5.0) | 10 (28.6) | 6 (25.0) | 0 |
| Nationality, n (%) | | | | | | |
| - Thai | 176 (91.2) | 7 (70.0) | 101 (93.5) | 37 (86.0) | 25 (96.2) | 6 (100) |
| - Non-Thai | 17 (8.8) | 3 (30.0) | 7 (6.5) | 6 (14.0) | 1 (3.8) | 0 |
| Type of infection, n (%) | | | | | | |
| - Imported case | 40 (20.7) | 5 (50.0) | 19 (17.6) | 12 (27.9) | 4 (15.4) | 0 |
| - Local transmission case | 153 (79.3) | 5 (50.0) | 89 (82.4) | 31 (72.1) | 22 (84.6) | 6 (100) |
| Transmission link, n (%) | | | | | | |
| - Contact with a confirm case | 67 (34.7) | 4 (40.0) | 47 (43.5) | 12 (27.9) | 4 (15.4) | 0 |
| - Arrived from a country with widespread transmission of COVID-19 within 14 days before onset of illness | 40 (20.7) | 5 (50.0) | 19 (17.6) | 12 (27.9) | 4 (15.4) | 0 |
| - Attended or worked at a crowded place | 33 (17.1) | 0 | 21 (19.4) | 6 (14.0) | 4 (15.4) | 2 (33.3) |
| - Boxing stadium clusters | 44 (22.8) | 1 (10.0) | 17 (15.7) | 12 (27.9) | 11 (42.3) | 3 (50.0) |
| - Healthcare facility | 1 (0.5) | 0 | 0 | 0 | 1 (3.8) | 0 |
| - Unknown | 8 (4.1) | 0 | 4 93.7) | 1 (2.3) | 6 (7.7) | 1 (16.7) |
| Smoking (n = 157), n (%) | | | | | | |
| - Never | 128 (81.5) | 8 (88.9) | 66 (80.5) | 32 (80.0) | 19 (86.4) | 3 (75.0) |
| - Ever | 29 (18.5) | 1 (11.1) | 16 (19.5) | 8 (20.0) | 3 (13.6) | 1 (25.0) |
| Alcohol use (n = 164), n (%) | | | | | | |
| - No | 112 (68.3) | 5 (62.5) | 55 (64.7) | 28 (68.3) | 21 (84.0) | 3 (60.0) |
| - Yes | 52 (31.7) | 3 (37.5) | 30 (35.3) | 13 (31.7) | 4 (16.0) | 2 (40.0) |
| Coexisting conditions, n (%) | | | | | | |
| - Any* | 48 (24.9) | 3 (30.0) | 10 (9.3) | 16 (37.2) | 15 (57.7) | 4 (66.7) |
| - Diabetes | 16 (8.3) | 1 (10.0) | 3 (2.8) | 2 (4.7) | 7 (26.9) | 3 (50.0) |
| - Hypertension | 31 (16.1) | 3 (30.0) | 4 (3.7) | 11 (25.6) | 10 (38.5) | 3 (50.0) |
| - Dyslipidemia | 10 (5.2) | 1 (10.0) | 2 (1.9) | 3 (7.0) | 3 (11.5) | 1 (16.7) |
| - Allergy | 2 (1.0) | 0 | 1 (0.9) | 0 | 1 (3.8) | 0 |
| - Chronic pulmonary diseases | 3 (1.6) | 0 | 1 (0.9) | 2 (4.7) | 0 | 0 |

*(Continued)*

**Table 1.** (Continued)

| | All (n = 193) | Asymptomatic (n = 10) | Mild (n = 108) | Moderate (n = 43) | Severe (n = 26) | Critical (n = 6) |
|---|---|---|---|---|---|---|
| - Chronic heart diseases | 2 (1.0) | 0 | 0 | 1 (2.3) | 1 (3.8) | 0 |
| - Chronic liver diseases | 5 (2.6) | 0 | 1 (0.9) | 3 (7.0) | 1 (3.8) | 0 |
| - HIV infection | 2 (1.0) | 0 | 1 (0.9) | 0 | 1 (3.8) | 0 |
| Angiotensin-converting enzyme inhibitors use, n (%) | 6 (3.1) | 0 | 3 (2.8) | 1 (2.3) | 1 (3.8) | 1 (16.7) |
| Angiotensin-receptor blockers use, n (%) | 11 (5.7) | 2 (20.0) | 1 (0.9) | 6 (14.0) | 2 (7.7) | 0 |
| Duration from onset of illness to the first visit, median (IQR), d | 3.0 (2.0–6.0) | - | 3.0 (2.0–6.0) | 3.0 (1.0–7.0) | 4.5 (1.0–6.3) | 4 (2.8–5.5) |
| Presenting symptoms*, n (%) | | | | | | |
| - Fever | 121 (62.7) | 0 | 60 (55.6) | 33 (76.7) | 23 (88.5) | 5 (83.3) |
| - Dry cough | 95 (49.2) | 0 | 50 (46.8) | 25 (58.1) | 18 (69.2) | 2 (33.3) |
| - Productive cough | 41 (21.2) | 0 | 22 (20.4) | 8 (18.6) | 7 (26.9) | 4 (67.7) |
| - Shortness of breath | 25 (13.0) | 0 | 8 (7.4) | 8 (18.6) | 8 (30.8) | 1 (16.7) |
| - Sore throat | 54 (28.0) | 0 | 42 (38.9) | 10 (23.3) | 1 (3.8) | 1 (16.7) |
| - Rhinorrhea | 55 (28.5) | 0 | 41 (38.0) | 10 (23.3) | 4 (15.4) | 0 |
| - Fatigue | 30 (15.5) | 0 | 15 (13.9) | 7 (16.3) | 8 (30.8) | 0 |
| - Myalgia/body aches | 69 (35.8) | 0 | 32 (29.6) | 20 (46.5) | 12 (46.2) | 5 (83.5) |
| - Headache | 25 (13.0) | 0 | 18 (16.3) | 4 (9.3) | 0 | 1 (16.7) |
| - Diarrhea | 15 (7.8) | 0 | 9 (8.3) | 1 (2.3) | 1 (3.8) | 1 (16.7) |
| - Poor appetite | 4 (2.1) | 0 | 1 (0.9) | 1 (2.3) | 2 (7.7) | 0 |
| - Nausea or vomiting | 5 (2.6) | 0 | 1 (0.9) | 1 (2.3) | 3 (11.5) | 0 |
| - Reduced sense of taste | 8 (4.1) | 0 | 3 (2.8) | 1 (2.3) | 4 (15.4) | 0 |
| - Reduced sense of smell | 11 (5.7) | 0 | 7 (6.5) | 1 (2.3) | 3 (11.5) | 0 |
| - No symptoms | 13 (6.7) | 10 (100) | 3 (2.8) | 0 | 0 | 0 |
| **Vital signs at the first presentation, initial laboratory and radiographic findings** | | | | | | |
| Body temperature, mean (±SD),˚C | 37.3 (0.8) | 36.6 (0.3) | 37.0 (0.6) | 37.5 (1.0) | 37.8 (1.1) | 38.0 (0.8) |
| Body temperature range distribution (n = 191), n (%) | | | | | | |
| <37.3˚C | 115 (60.2) | 10 (100) | 77 (72.6) | 19 (44.2) | 9 (34.6) | 0 |
| 37.3–38.0˚C | 46 (24.1) | 0 | 24 (22.6) | 12 (27.9) | 6 (22.3) | 4 (66.7) |
| 38.1–39.0˚C | 20 (10.5) | 0 | 4 (3.8) | 9 (20.9) | 6 (22.3) | 1 (16.7) |
| >39.0˚C | 10 (5.2) | 0 | 1 (0.9) | 3 (7.0) | 5 (19.2) | 1 (16.7) |
| Respiratory rate, median (IQR), breaths/min | 18 (18–20) | 18 (18–19) | 18 (18–19) | 18 (18–19) | 20 (18–22) | 20 (19–28) |
| Oxygen saturation at presentation, median (IQR), % | 98 (97–99) | 99 (98–99) | 99.0 (98–100) | 98 (97–99) | 97 (95–98) | 96 (88–98) |
| Initial laboratory findings | | | | | | |
| White blood cell count, median (IQR), x10$^9$ /L | 5.9 (4.6–7.1) | 6.8 (5.8–7.7) | 5.9 (4.7–7.3) | 5.3 (4.1–6.2) | 6.2 (5.4–8.0) | 6.9 (4.8–8.4) |
| Absolute neutrophil count, median (IQR), x10$^9$ /L | 3.5 (2.6–4.9) | 3.6 (2.7–5.1) | 3.5 (2.7–4.8) | 3.0 (2.2–4.0) | 4.7 (3.2–6.3) | 5.8 (4.1–6.5) |
| Absolute lymphocyte count, median (IQR), x10$^9$ /L | 1.6 (1.1–2.1) | 2.0 (1.4–2.3) | 1.8 (1.3–2.2) | 1.4 (1.1–1.9) | 1.3 (0.9–1.5) | 0.8 (0.5–0.9) |
| Absolute monocyte count, median (IQR), x10$^9$ /L | 0.3 (0.2–0.5) | 0.5 (0.3–0.5) | 0.4 (0.2–0.5) | 0.3 (0.2–0.5) | 0.3 (0.2–0.4) | 0.3 (0.2–0.4) |
| Hemoglobin, median (IQR), g/dL | 13.6 (12.6–14.6) | 13.5 (12.2–13.7) | 13.3 (12.5–14.2) | 13.9 (12.9–14.9) | 14.1 (12.3–15.0) | 13.6 (12.1–14.3) |
| Platelet count, median (IQR), x10$^9$ /L | 221 (181–280) | 243 (207–276) | 240 (195–300) | 199 (164–226) | 204 (156–260) | 165 (162–188) |
| Sodium level (n = 96), median (IQR), mEq/L, | 139 (137–141) | 141 (139–143) | 140 (139–141) | 139 (137–141) | 137 (133–139) | 136 (133–141) |
| Potassium level (n = 96), median (IQR), mEq/L, | 3.9 (3.6–4.2) | 3.8 (2.9–3.9) | 4.1 (3.9–4.3) | 3.9 (3.6–4.1) | 3.6 (3.4–4.3) | 3.8 (3.2–4.0) |
| Chlorine level (n = 96), median (IQR), mEq/L | 102 (99–103) | 102 (99–103) | 102 (101–104) | 102 (100–103) | 98 (95–102) | 102 (96–103) |
| Bicarbonate level (n = 96), median (IQR), mEq/L | 24 (23–25) | 25 (23–27) | 24 (22–25) | 24 (23–25) | 24 (21–25) | 23 (18–26) |

(Continued)

**Table 1.** (Continued)

|  | All (n = 193) | Asymptomatic (n = 10) | Mild (n = 108) | Moderate (n = 43) | Severe (n = 26) | Critical (n = 6) |
|---|---|---|---|---|---|---|
| Creatinine, median (n = 112), mg/dL, | 0.8 (0.7–1.0) | 0.8 (0.6–0.9) | 0.7 (0.6–0.9) | 0.9 (0.7–1.0) | 0.8 (0.7–1.1) | 1.1 (1.0–1.3) |
| Aspartate aminotransferase, (n = 104), median (IQR), U/L, | 24 (19–35) | 21 (18–31) | 21 (17–25) | 28 (22–41) | 33 (23–39) | 78 (52–85) |
| Alanine aminotransferase (n = 104), median (IQR), U/L | 22 (15–33) | 22 (16–23) | 18 (12–25) | 27 (20–41) | 21 (14–38) | 48 (41–64) |
| Rapid influenza diagnosis test, n (%) |  |  |  |  |  |  |
| - Negative | 140 (72.5) | 6 (60.0) | 74 (68.5) | 37 (86.0) | 19 (73.1) | 4 (66.7) |
| - Positive | 1 (0.5) | 0 | 1 (0.9) | 0 | 0 | 0 |
| - Not tested | 52 (26.9) | 4 (60.0) | 33 (30.6) | 6 (14.0) | 7 (26.9) | 2 (33.3) |
| Initial chest film opacities, n (%) |  |  |  |  |  |  |
| None | 156 (80.8) | 10 (100) | 108 (100) | 22 (51.2) | 12 (46.2) | 4 (66.7) |
| Unilateral | 17 (8.8) | 0 | 0 | 13 (30.2) | 4 (15.4) | 0 |
| Bilateral | 20 (10.4) | 0 | 0 | 8 (18.6) | 10 (38.5) | 2 (33.3) |

Abbreviations: BMI, body mass index; HIV, human immunodeficiency virus.

* More than one pre-existing condition or presenting symptoms could be given for these characteristics

absolute lymphocyte and platelet count). Rapid influenza diagnosis test was done in 141 (73.0%) patients; only one patient tested positive for influenza A. Chest radiography revealed no opacity in 80.8%, unilateral opacity in 8.8%, and bilateral opacities in 10.4% of the patients.

## During hospitalization

The median (IQR) time from onset of illness to hospitalization was 5.0 (3.0–7.0) days. The median frequency of chest radiograph performed during admission was 3.0 (2.0–5.0). Follow-up chest radiograph was done in 155 (80.3%) of the patients. Pneumonia was detected in 75 (38.9%) of the patients, of which 49.3% were upon admission and 50.7% were during hospitalization. Among 38 patients who progressed to have pneumonia after admission, diagnostic work up to rule out hospital-acquired pneumonia was performed. Only one patient had *Hemophilus parainfluenzae* detected from sputum culture which was considered as co-infection. Among the 75 cases with pneumonia, 34.7% were unilateral, and 65.3% were bilateral. The median (IQR) time from onset of illness to pneumonia detection was 7.0 (5.0–9.0) days. Fever was present in only 49.2% of all cases but was detected in 88% of patients with pneumonia. Among febrile patients, non-pneumonia (mild) cases had a lower mean of highest temperature during hospitalization than of those who had pneumonia (37.9 vs 38.8°C). Of 121 patients who reported having had subjective fever prior to admission, 44 (36.4%) had no fever for the entire length of hospital stay. Of 72 patients who reported no fever prior to admission, 18 (25.0%) developed a fever during hospitalization. The median duration from admission to defervescence was 5.0 (3.0–9.0) days. The median duration from admission to defervescence in mild cases was the shortest (3.0 days) compared to higher severity disease categories (Table 2). Seventy-four patients (38.3%) received supportive treatment while 61.7% also received therapeutic options listed in the Thai treatment guideline for cases of COVID-19 infection (Table 2). Thirty-two patients (16.6%) were transferred to the intensive care unit (ICU). The median (IQR) duration from illness onset to ICU admission was 8.0 (5.3–10.0) days. Oxygen saturation of < 95% was found in 18.1% of cases whereas 13.0% experienced a respiratory rate of ≥ 24 breaths/min, which only occurred in the pneumonia group. The median duration (IQR) from symptom onset to oxygen saturation < 95% and respiratory rate ≥ 24 breaths/min were 8.0 (7.0–9.0) and 9.0 (6.0–11.0) days, respectively. Supplemental

**Table 2. Treatments and clinical course during hospitalization.**

| | All (n = 193) | Asymptomatic (n = 10) | Mild (n = 108) | Moderate (n = 43) | Severe (n = 26) | Critical (n = 6) |
|---|---|---|---|---|---|---|
| **Treatments** | | | | | | |
| Supportive, n (%) | 74 (38.3) | 9 (90.0) | 53 (49.1) | 10 (23.3) | 2 (7.7) | 0 |
| Chloroquine monotherapy, n (%) | 20 (10.4) | 0 | 20 (18.5) | 0 | 0 | 0 |
| Chloroquine or hydroxychloroquine + boosted lopinavir or darunavir, n (%) | 36 (18.7) | 1 (10.0) | 28 (25.9) | 1 (2.3) | 4 (15.4) | 2 (33.3) |
| Hydroxychloroquine + azithromycin, n (%) | 8 (4.1) | 0 | 7 (6.5) | 1 (2.3) | 0 | 0 |
| Chloroquine or hydroxychloroquine + boosted lopinavir or darunavir + azithromycin, n (%) | 5 (2.6) | 0 | 0 | 1 (2.3) | 3 (11.5) | 1 (16.7) |
| Chloroquine or hydroxychloroquine + boosted lopinavir or darunavir + favipiravir, n (%) | 38 (19.7) | 0 | 0 | 27 (62.8) | 10 (38.5) | 1 (16.7) |
| Chloroquine or hydroxychloroquine + boosted lopinavir or darunavir + azithromycin + favipiravir, n (%) | 12 (6.2) | 0 | 0 | 3 (7.0) | 7 (26.9) | 2 (33.3) |
| Remdesivir, n (%) | 7 (3.6) | 0 | 0 | 0 | 5 (19.2) | 2 (33.3) |
| Tocilizumab, n (%) | 3 (1.6) | 0 | 0 | 0 | 2 (8.0) | 1 (16.7) |
| Coticosteriod, n (%) | 2 (1.0) | 0 | 0 | 0 | 0 | 2 (33.3) |
| Convalescent plasma, n (%) | 2 (1.0) | 0 | 0 | 0 | 0 | 2 (33.3) |
| Antibiotics, n (%) | 27 (14.0) | 0 | 3 (2.8) | 7 (16.3) | 12 (46.2) | 5 (83.3) |
| Oxygen therapy, n (%) | | | | | | |
| - Nasal cannula/face mask | 36 (18.7) | 0 | 0 | 4 (9.3) | 26 (100) | 6 (100) |
| - High-flow oxygen | 9 (4.7) | 0 | 0 | 0 | 5 (19.2) | 4 (66.7) |
| - Invasive ventilation | 5 (2.6) | 0 | 0 | 0 | 0 | 5 (83.3) |
| Duration of oxygen therapy, median (IQR), d | 5.0 (2.5–11.0) | - | - | 2.0 (1.3–2.8) | 5.5 (3.0–11.0) | > 18.0 |
| Extracorporeal membrane oxygenation, n (%) | 1 (0.51) | 0 | 0 | 0 | 0 | 1 (16.7) |
| Continuous renal replacement therapies, n (%) | 1 (0.51) | 0 | 0 | 0 | 0 | 1 (16.7) |
| **Clinical course, complications and final clinical outcomes** | | | | | | |
| Duration from onset of illness to admission, median (IQR), d | 5.0 (3.0–7.0) | - | 5.0 (3.0–7.0) | 5.0 (3.0–9.0) | 5.5 (4.0–9.0) | 5.5 (2.8–7.0) |
| Fever during hospitalization, n (%) | 95 (49.2) | 0 | 29 (26.9) | 35 (81.4) | 25 (96.2) | 6 (100) |
| Highest temperature during hospitalization, mean (±SD), ˚C | 38.5 (0.8) | < 37.3 | 37.9 (0.5) | 38.5 (0.6) | 39.0 (0.8) | 39.5 (0.9) |
| - Duration from admission to defervescence, median (IQR), d | 5.0 (3.0–9.0) | - | 3.0 (1.0–5.5) | 6.0 (3.0–9.0) | 7.0 (5.0–11.5) | 15.0 |
| Worst opacity in chest film, n (%) | | | | | | |
| None | 118 (61.1) | 10 (100) | 108 (100) | 0 | 0 | 0 |
| Unilateral | 26 (13.5) | 0 | 0 | 24 (55.8) | 2 (7.7) | 0 |
| Bilateral | 49 (25.4) | 0 | 0 | 19 (44.2) | 24 (92.3) | 6 (100) |
| Oxygen saturation < 95%, n (%) | 35 (18.1) | 0 | 0 | 3 (7.0) | 26 (100) | 6 (100) |
| Respiratory rate ≥ 24 breaths/min, n (%) | 25 (13.0) | 0 | 0 | 2 (4.7) | 17 (65.4) | 6 (100) |
| ICU admission, n (%) | 32 (16.6) | 0 | 0 | 6 (14.0) | 20 (76.9) | 6 (100) |
| ARDS, n (%) | 6 (3.1) | 0 | 0 | 0 | 0 | 6 (100) |
| Acute kidney injury, n (%) | 7 (3.6) | 0 | 0 | 2 (4.7) | 0 | 5 (83.3) |
| Co-infection*, n (%) | 8 (4.1) | 0 | 3 (2.8) | 2 (4.7) | 1 (3.7) | 2 (33.3) |
| Length of hospital stay, median (IQR), d | 12.0 (7.5–19.0) | 8.5 (5.8–20.8) | 10.5 (7.0–16.0) | 13.0 (9.0–18.0) | 16.0 (12.0–22.3) | 32.5 (19.3–51.2) |
| Duration of viral RNA shedding after onset of illness, median (IQR), d | 16.0 (11.0–24.0) | 6.0 (4.8–19.0) | 13.0 (9.0–21.0) | 16.0 (12.0–24.0) | 20.5 (13.0–24.0) | 26.5 (21.5–34.5) |
| Final clinical outcome, n (%) | | | | | | |
| - Recovered | 189 (97.9) | 10 (100) | 108 (100) | 43 (100) | 26 (100) | 2 (33.3) |
| - Deceased | 4 (2.1) | 0 | 0 | 0 | 0 | 4 (66.7) |

(*Continued*)

**Table 2.** (Continued)

|  | All (n = 193) | Asymptomatic (n = 10) | Mild (n = 108) | Moderate (n = 43) | Severe (n = 26) | Critical (n = 6) |
|---|---|---|---|---|---|---|
| - Remained hospitalized, n (%) | 0 | 0 | 0 | 0 | 0 | 0 |

Abbreviations: ARDS, acute respiratory distress syndrome; ICU, intensive care unit.

* Pulmonary tuberculosis in 2, *H. influenza*e in 2, Influenza A in 1, adenovirus in 1, *H. parainfluenzae* in 1, *K. pneumoniae* in 1

oxygen by nasal cannula or face mask was administered in 18.7%, high-flow oxygen in 4.7% and mechanical ventilation in 2.6% of all the patients. The median duration (IQR) from illness onset to intubation was 9.0 (7.0–12.5) days. The median (IQR) duration of oxygen therapy was 5.0 (2.5–11.0) days. More than two weeks of oxygen therapy was required in critical cases. Moderate to severe ARDS was found in 3.1% whereas 3.6% of the patients developed AKI. The median (IQR) length of hospital stay was 12.0 (7.5–19.0) days. The median duration of viral RNA shedding after the onset of symptom was 16.0 (11.0–24.0) days. Severe cases had a longer viral shedding duration than the non-severe cases (Table 2). The longest observed duration of viral shedding was 45 days.

## Final clinical outcomes and factors associated with pneumonia

Of all cases, 189 (97.9%) were recovered and discharged whereas 4 (2.1%) were deceased. The degree of disease severity was classified as asymptomatic in 5.2%, mild in 55.9%, moderate (non-severe pneumonia) in 22.3%, severe (severe pneumonia) in 13.5%, and critical in 3.1% (Fig 1). According to the Chinese CDC definition [15], 83.4% of the patients were considered mild, 13.5% were severe, and 3.1% were critical. The median time (IQR) from the onset of ill-ness to death was 30.0 (18.0–49.5) days.

Table 3 demonstrates that patients with pneumonia were older (p<0.001), more likely to be male (p = 0.001), more likely to be obese (p = 0.001), and were different in many presenting

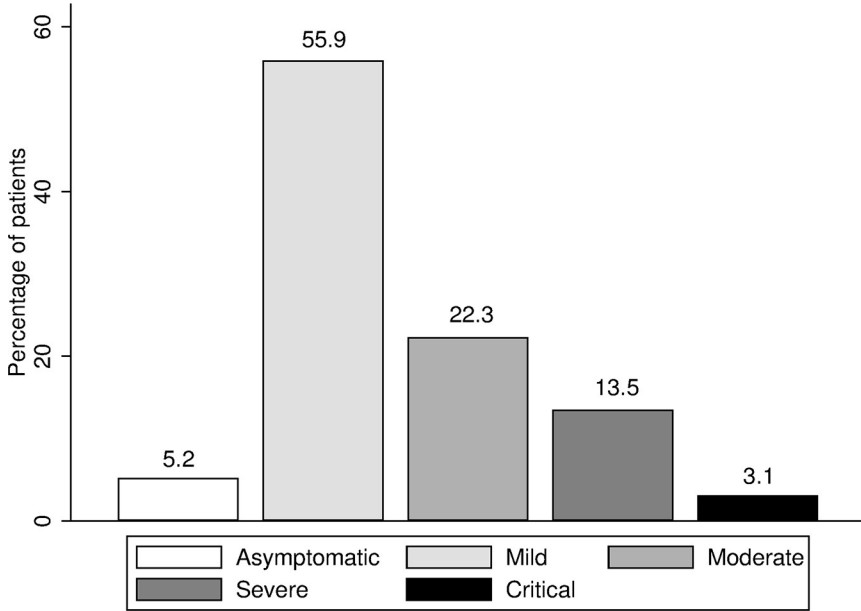

**Fig 1. Disease severity classification.**

**Table 3.  Baseline characteristics, initial laboratory findings, and outcomes of patients with COVID-19 comparing the pneumonia group to the non-pneumonia group (n = 193).**

| | Non-pneumonia (n = 118) | Pneumonia (n = 75) | p-value |
|---|---|---|---|
| **Baseline characteristics** | | | |
| Age, median (IQR), y | 33.0 (26.0–42.3) | 51.0 (38.0–59.0) | <0.001 |
| Gender, n (%) | | | 0.001 |
| - Male | 58 (49.2) | 55 (73.3) | |
| - Female | 60 (50.8) | 20 (26.7) | |
| BMI, median (IQR), kg/m$^2$ | 21.8 (19.4–24.7) | 25.1 (22.8–29.9) | <0.001 |
| - Obesity (BMI ≥30), n (%) | 6 (5.1) | 16 (21.3) | 0.001 |
| Nationality, n (%) | | | 0.934 |
| - Thai | 108 (91.5) | 68 (90.7) | |
| - Non-Thai | 10 (8.5) | 7 (9.3) | |
| Type of infection, n (%) | | | 0.853 |
| - Imported cases | 24 (20.3) | 16 (21.3) | |
| - Local transmission cases | 94 (79.7) | 59 (78.7) | |
| Transmission link, n (%) | | | 0.007 |
| - Contact with a confirm case | 51 (43.2) | 16 (21.3) | |
| - Travel history within 14 days before onset of symptom | 24 (20.3) | 16 (21.3) | |
| - Attended or worked at crowded places | 21 (17.8) | 12 (16.0) | |
| - Boxing stadium clusters | 18 (15.3) | 26 (34.7) | |
| - Healthcare facility | 0 | 1 (1.3) | |
| - Unknown | 4 (3.8) | 4 (6.3) | |
| Smoking (n = 157), n (%) | | | 1.000 |
| - Ever | 17 (18.7) | 12 (18.2) | |
| Alcohol use (n = 164), n (%) | | | 0.310 |
| - Yes | 33 (35.5) | 19 (26.8) | |
| Coexisting conditions, n (%) | | | |
| - Any | 13 (11.0) | 35 (46.7) | <0.001 |
| - Diabetes | 4 (3.4) | 12 (16.0) | 0.003 |
| - Hypertension | 7 (5.9) | 24 (32.0) | <0.001 |
| - Dyslipidemia | 3 (2.5) | 7 (9.3) | 0.050 |
| - Allergy | 1 (0.8) | 1 (1.3) | 1.000 |
| - Chronic pulmonary diseases | 1 (0.8) | 2 (2.7) | 0.561 |
| - Chronic heart diseases | 0 | 2 (2.7) | 0.150 |
| - Chronic liver diseases | 1 (0.8) | 4 (5.3) | 0.076 |
| - HIV infection | 1 (0.8) | 1 (1.3) | 1.000 |
| Angiotensin-converting enzyme inhibitors use, n (%) | 3 (2.5) | 3 (4.0) | 0.679 |
| Angiotensin-receptor blockers use, n (%) | 3 (2.5) | 8 (10.7) | 0.025 |
| Presenting symptoms, n (%) | | | |
| - Fever | 60 (50.8) | 61 (81.3) | <0.001 |
| - Dry cough | 50 (42.4) | 45 (60.0) | 0.019 |
| - Productive cough | 22 (18.6) | 19 (25.3) | 0.283 |
| - Shortness of breath | 6 (6.8) | 17 (22.7) | 0.002 |
| - Sore throat | 42 (35.6) | 12 (16.0) | 0.003 |
| - Rhinorrhea | 41 (34.7) | 14 (18.7) | 0.022 |
| - Fatigue | 15 (12.7) | 15 (20.0) | 0.221 |
| - Myalgia/body aches | 32 (27.1) | 37 (49.3) | 0.002 |
| - Headache | 18 (15.3) | 5 (6.7) | 0.276 |

*(Continued)*

**Table 3.** (Continued)

|  | Non-pneumonia (n = 118) | Pneumonia (n = 75) | p-value |
|---|---|---|---|
| - Diarrhea | 9 (7.6) | 6 (8.0) | 1.000 |
| - Poor appetite | 1 (0.8) | 3 (4.0) | 0.301 |
| - Nausea or vomiting | 1 (0.8) | 4 (5.3) | 0.076 |
| - Reduced sense of taste | 3 (2.5) | 5 (6.7) | 0.265 |
| - Reduced sense of smell | 7 (5.9) | 4 (5.3) | 1.000 |
| - No symptoms | 13 (11.0) | 0 | 0.002 |
| Body temperature at presentation, mean (±SD),˚C | 37.0 (0.6) | 37.6 (1.0) | <0.001 |
| Respiratory rate at presentation, median (IQR), breaths/min | 18 (18–20) | 20 (18–20) | 0.001 |
| Oxygen saturation at presentation, median (IQR), % | 99 (98–100) | 98 (97–99) | <0.001 |
| **Initial laboratory findings** |  |  |  |
| White blood cell count, median (IQR), $x10^9$ /L | 5.9 (4.8–7.3) | 5.8 (4.3–6.9) | 0.298 |
| Absolute neutrophil count, median (IQR), $x10^9$ /L | 3.5 (2.7–4.8) | 3.5 (2.6–5.2) | 0.580 |
| Absolute lymphocyte count, median (IQR), $x10^9$ /L | 1.8 (1.3–2.2) | 1.3 (0.9–1.7) | <0.001 |
| Absolute monocyte count, median (IQR), $x10^9$ /L | 0.4 (0.3–0.5) | 0.3 (0.2–0.5) | 0.413 |
| Hemoglobin, median (IQR), g/dL | 13.3 (12.6–14.0) | 14.0 (12.5–15.0) | 0.061 |
| Platelet count, median (IQR), $x10^9$ /L | 240 (194–247) | 194 (157–226) | <0.001 |
| Sodium level, median (IQR), mEq/L | 140 (139–141) | 138 (136–140) | 0.004 |
| Potassium level, median (IQR), mEq/L | 4.0 (3.8–4.2) | 3.8 (3.5–4.1) | 0.005 |
| Chlorine level, median (IQR), mEq/L | 102 (101–104) | 100 (97–103) | 0.005 |
| Bicarbonate level, median (IQR), mEq/L | 24 (23–25) | 24 (23–25) | 0.435 |
| Creatinine, median (IQR), mg/dL | 0.7 (0.8–0.9) | 0.9 (0.7–1.1) | <0.001 |
| Aspartate aminotransferase, median (IQR), U/L | 21 (18–25) | 31 (23–43) | <0.001 |
| Alanine aminotransferase, median (IQR), U/L | 19 (13–25) | 26 (19–42) | 0.001 |
| **Outcomes** |  |  |  |
| Fever during hospitalization, n (%) | 29 (24.6) | 66 (88.0) | <0.001 |
| Highest temperature during hospitalization, mean (±SD),˚C | 37.9 (0.5) | 38.8 (0.8) | <0.001 |
| Duration from admission to defervescence, median (IQR), d | 3.0 (1.0–5.5) | 6.0 (4.0–10.0) | <0.001 |
| ICU admission n, (%) | 0 | 32 (42.7) | <0.001 |
| Length of hospital stay, median (IQR), d | 10.0 (6.8–16.0) | 14.0 (10–23.0) | <0.001 |
| Duration of viral RNA shedding after onset of symptom, median (IQR), d | 14.0 (10–24.0) | 18.0 (13.0–24.0) | 0.023 |
| Final clinical outcomes |  |  | 0.002 |
| Recovered, n (%) | 118 (100) | 71 (94.7) |  |
| Deceased, n (%) | 0 | 4 (5.3) |  |

Abbreviations: BMI, body mass index; HIV, human immunodeficiency virus

symptoms (p<0.05) than those without pneumonia. Cases with pneumonia also had more comorbidities (p<0.001), higher body temperature (p<0.001), and lower oxygen saturation at presentation (p<0.001) than the non-pneumonia cases. Patients without pneumonia more frequently complained of a runny nose (p = 0.022) and sore throat (p = 0.003) than those with pneumonia. Cases with pneumonia also had a higher proportion of febrile illness during hospitalization (p<0.001), longer duration from admission to defervescence (p<0.001), longer hospital stay (p<0.001), and longer viral shedding duration (p<0.001) than patients without pneumonia.

As summarized in Table 4, age (OR 2.55 per 10-year increase from 30 years old; 95% CI, 1.67–3.90; p<0.001), obesity (OR 8.74; 95% CI, 2.06–37.18; p = 0.003), and body temperature at presentation (OR 4.59 per 1˚C increase from 37.2˚C; 95% CI, 2.30–9.17; p<0.001) were significantly associated with COVID-19 pneumonia.

**Table 4. Logistic regression analysis of factors associated with COVID-19 pneumonia.**

| | Crude OR (95%CI) | p-value | Adjusted (95%CI) | p-value |
|---|---|---|---|---|
| Gender | | | | |
| Female | 1 (reference) | | | |
| Male | 2.85 (1.52–5.32) | 0.001 | 2.28 (0.79–6.56) | 0.128 |
| Age, for every 10-year increase from 30 years old | 2.24 (1.73–2.90) | <0.001 | 2.55 (1.67–3.90) | <0.001 |
| Body mass index (BMI) | 1.20 (1.11–1.30) | <0.001 | | |
| - BMI < 30 kg/m$^2$ | 1 (reference) | | | |
| - BMI ≥ 30 kg/m$^2$ (obesity) | 5.55 (2.05–15.06) | 0.001 | 8.74 (2.06–37.18) | 0.003 |
| Nationality | | | | |
| - Non-Thai | 1 (reference) | | | |
| - Thai | 1.26 (0.11–14.16) | 0.825 | | |
| Type of infection | | | | |
| - Imported case | 1 (reference) | | | |
| - Local transmission case | 0.94 (0.46–1.92) | 0.868 | | |
| Transmission link | | | | |
| - Contact with a confirm case | 1 (reference) | | | |
| - Travel history within 14 days before onset of symptom | 2.13 (0.91–4.95) | 0.081 | | |
| - Attended or worked at crowded places | 1.82 (0.74–4.50) | 0.194 | | |
| - Boxing stadium clusters | 4.60 (2.02–10.48) | <0.001 | 1.02 (0.31–3.34) | 0.968 |
| - Unknown | 3.19 (0.72–14.22) | 0.129 | | |
| Smoking (vs never) | 0.98 (0.43–2.19) | 0.937 | | |
| Current alcohol use (vs no use) | 0.66 (0.34–1.31) | 0.235 | | |
| Coexisting conditions | | | | |
| - Diabetes | 5.43 (1.48–17.54) | 0.005 | 1.12 (0.22–5.88) | 0.890 |
| - Hypertension | 7.46 (3.02–18.44) | <0.001 | 1.08 (0.24–4.94) | 0.925 |
| - Dyslipidemia | 3.95 (0.98–15.77) | 0.052 | | |
| - Allergy | 1.58 (0.10–25.67) | 0.741 | | |
| - Chronic pulmonary diseases | 3.21 (0.29–35.98) | 0.345 | | |
| - Chronic liver diseases | 6.60 (0.72–60.15) | 0.095 | | |
| - HIV infection | 1.58 (0.10–25.66) | 0.747 | | |
| Angiotensin-converting enzyme inhibitors use | 1.60 (0.31–8.13) | 0.573 | | |
| Angiotensin-receptor blockers use* | 4.58 (1.17–17.85) | 0.028 | | |
| Body temperature at presentation, per 1˚C increase from 37.2˚C | 3.55 (2.23–5.64) | <0.001 | 4.59 (2.30–9.17) | <0.001 |
| White blood cell count, x10$^9$ /L | 0.93 (0.79–1.08) | 0.311 | | |
| Absolute neutrophil count, x10$^9$ /L | 1.06 (0.90–1.25) | 0.512 | | |
| Absolute lymphocyte count, x10$^9$ /L | 0.28 (0.16–0.48) | <0.001 | | |
| - Absolute lymphocyte < 1,500 per mm$^3$ | 3.69 (2.00–6.82) | <0.001 | 1.73 (0.65–4.62) | 0.276 |
| Absolute monocyte count, x10$^9$ /L | 0.47 (0.09–2.48) | 0.376 | | |
| Hemoglobin level, g/dL | 1.19 (0.99–1.44) | 0.07 | | |
| Platelet count, x10$^9$ /L | 0.99 (0.98–0.99) | <0.001 | | |
| - Platelet count < 150 per mm$^3$ | 9.84 (2.11–45.86) | 0.004 | 4.03 (0.53–30.83) | 0.169 |

Abbreviations: BMI, body mass index; CI, confidence interval; HIV, human immunodeficiency virus; OR, odds ratio.

* Angiotensin-receptor blockers use significantly correlated with hypertension (r = 0.56, p<0.001)

## Discussion

We describe the clinical spectrum and outcomes of 193 COVID-19 patients admitted at a national infectious institute in Thailand. More than half of the patients had a mild disease severity and the recovery rate of our cohort was 97.9% with a case fatality rate of 2.1%—only

four deaths were observed in six critical patients. The overall incidence of pneumonia was 38.9%, of which 57.3% were not severe. Increasing age, obesity, and higher body temperature were potential predictive factors for pneumonia in SARS-CoV-2 infected patients.

Our cohort included laboratory-confirmed COVID-19 patients hospitalized regardless of their disease severity. Findings from our study could be representative of the patients in the full spectrum of the disease from the first presentation through to the final clinical outcomes. A wide range of mortality of COVID-19 patients from 0 to 28% has been reported in previous studies [3–5, 8, 9, 11, 12, 24], which may have resulted from selection bias of either mild disease status or severe disease status along with a short observation period.

In the present study, approximately 40% of patients with SARS-CoV2 infection developed pneumonia. This was much lower than incidence of pneumonia from SARS-CoV-1 (78–90%) [25, 26]. We also found bilateral pneumonia was more prevalent than unilateral pneumonia, which was different from Severe Acute Respiratory Syndrome (SARS) [27]. Older age is widely recognized to be associated with worse pneumonia [11, 28–30]. However, obesity has been less explored so far. Obesity can impair immune responses to viral infection [31, 32]. Kass, *et al.* found younger individuals with COVID-19 admitted to hospital were more likely to be obese [33]. Chen, *et al.* reported those with obesity were more likely to have severe condition [30]. We identified obesity was significantly associated with COVID-19 pneumonia.

We found that fever was not a hallmark of COVID-19 but fever on admission was significantly associated with pneumonia in the multivariate analysis. Although fever was the most common presenting symptom, presenting fever on admission was less common in patients with mild disease than those with moderate to severe disease. Among cases with mild severity, only 29 (48.3%) of 60 cases who reported fever had a fever during hospitalization. With a median duration from illness onset to the admission of five days, half of the febrile patients with mild COVID-19 might have had a fever for less than five days. Furthermore, we observed that patients with mild disease more frequently had sore throat and rhinorrhea compared with those who were moderate to severe disease severities. These may indicate that the virus is limited to the upper respiratory tract in patients with mild disease. Of note, some patients with mild disease had advanced age and obesity.

Few studies have reported on the proportion of asymptomatic infection. Our study revealed 13 patients who were asymptomatic at presentation, but three of them subsequently developed symptoms and were recategorized as presymptomatic. Hence, the asymptomatic infection was estimated to be 5% in our cohort, which differs from the previously published reports in other settings (17.9–42.3%) [34, 35]. More information on the actual incidence of asymptomatic infection among SARS-CoV-2 infected patients is needed. Interestingly, 30% of asymptomatic cases in our cohort were more than 50 years of age. Determinants of disease severity among the elderly required further investigation.

We did not intend to demonstrate the efficacy of a specific treatment on COVID-19. Patients with mild to moderate disease received only supportive care and recovered. They could be discharged, suggesting the self-limiting nature of the non-severe cases. While published randomized trials on chloroquine, hydroxychloroquine, and lopinavir/ritonavir have been unable to demonstrate treatment benefit [36–38], supportive care is crucial for COVID-19 patients with the mild or moderate disease.

Understanding the full spectrum of COVID-19 is essential for estimating the proportion of severe COVID-19 cases that require a large amount of healthcare resources. Demand for hospital inpatient and ICU beds could be better predicted to mitigate the overwhelming hospital burden after easing COVID-19 restriction. Although individuals without risk factors who present with mild disease generally do not require hospitalization, some of them might subsequently deteriorate.

This study has several limitations. First, the findings were based on a relatively small sample size from a single center and may not be generalizable to other settings. However, the proportion of patients in each category of the disease spectrum was comparable with those of nationwide survey of China. This suggests that our sample may be representative of patients with COVID-19 throughout the disease spectrum. Second, the study had risk of recall bias as patients were asked to recall subjective events prior to admission. Third, not all blood chemistry studies were performed in all patients and several non-routine tests (eg, serum LDH, C-reactive protein, IL-6 level) were not investigated. Fourth, we used chest radiograph as radiologic evidence of pneumonia. As chest radiograph is less sensitive than computed tomography, the very mild pneumonia might have been missed. Likewise, arterial blood gas was evaluated only in critical cases, so the incidence of ARDS may not have been correctly estimated. Lastly, our institute has no testing facility for the viral load of SARS-CoV-2 so the duration of viral RNA shedding may not represent the duration of viral viability.

In conclusion, the majority of patients with COVID-19 had mild illness. The incidence of pneumonia of any severity was 39% (non-severe in 22%, severe in 14%, critical in 3%). Most patients had a good final clinical outcomes. The case fatality rate in our cohort was 2.1%. Increasing age, obesity, and higher temperature at presentation were potential predictive factors of COVID-19 pneumonia.

## Acknowledgments

The authors would like to thank Aroon Luaengniyomkul, Patama Suttha, Krittaecho Siripassorn, Anuttra A Ratnarathon, and Suparat Khemnark for their invaluable support for this study. We also greatly appreciate all the healthcare personnel at BIDI for their efforts in providing the best care for our patients.

## Author Contributions

**Conceptualization:** Wannarat A. Pongpirul, Surasak Wiboonchutikul.

**Data curation:** Wannarat A. Pongpirul, Surasak Wiboonchutikul, Lantharita Charoenpong, Nayot Panitantum.

**Formal analysis:** Wannarat A. Pongpirul, Surasak Wiboonchutikul.

**Investigation:** Wannarat A. Pongpirul, Surasak Wiboonchutikul, Lantharita Charoenpong, Nayot Panitantum.

**Methodology:** Wannarat A. Pongpirul, Surasak Wiboonchutikul.

**Project administration:** Apichart Vachiraphan.

**Resources:** Wannarat A. Pongpirul, Surasak Wiboonchutikul, Lantharita Charoenpong, Nayot Panitantum, Sumonmal Uttayamakul.

**Supervision:** Weerawat Manosuthi, Wisit Prasithsirikul.

**Validation:** Wannarat A. Pongpirul, Surasak Wiboonchutikul.

**Visualization:** Wannarat A. Pongpirul, Surasak Wiboonchutikul.

**Writing – original draft:** Wannarat A. Pongpirul, Surasak Wiboonchutikul, Lantharita Charoenpong, Nayot Panitantum, Sumonmal Uttayamakul.

**Writing – review & editing:** Krit Pongpirul, Weerawat Manosuthi, Wisit Prasithsirikul.

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
