## [Decision Letter · Decision Letter 0]

18 Aug 2020

Dear Dr. Wiboonchutikul,

Thank you very much for submitting your manuscript "Clinical course and potential predicting factors of pneumonia of adult patients with coronavirus disease 2019 (COVID-19): A retrospective observational analysis of 193 confirmed cases in Thailand" for consideration at PLOS Neglected Tropical Diseases. As with all papers reviewed by the journal, your manuscript was reviewed by members of the editorial board and by several independent reviewers. The reviewers appreciated the attention to an important topic. Based on the reviews, we are likely to accept this manuscript for publication, providing that you modify the manuscript according to the review recommendations. 

Sincerely,

Johan Van Weyenbergh

Associate Editor

Victor Santos

Deputy Editor

Editor comments.

The manuscript will benefit from a grammatical review by a native speaker.

Reviewer's Responses to Questions

**Key Review Criteria Required for Acceptance?**

**Methods**

-Are the objectives of the study clearly articulated with a clear testable hypothesis stated?

-Is the study design appropriate to address the stated objectives?

-Is the population clearly described and appropriate for the hypothesis being tested?

-Is the sample size sufficient to ensure adequate power to address the hypothesis being tested?

-Were correct statistical analysis used to support conclusions?

-Are there concerns about ethical or regulatory requirements being met?

Reviewer #1: 1. In the study, initial chest film revealed opacity in 37 patients, but total numbers of pneumonia patients were 75. Pneumonia in this study was defined as the presence of respiratory symptoms and opacity on chest radiography. However, during admission, hospital-acquired pneumonia (HAP) from other pathogens can occur. How the authors excluded HAP from the progression of COVID-19? Also, the frequency of chest radiography to detect lung opacity and/or indication for chest film during hospital admission was not described.

2. ARB use might be collinearity with hypertension in multivariate logistic regression. This might result in no statistical significance of both factors. Some previous studies showed the association of hypertension and COVID-19 pneumonia or having adverse events from the infection.

3. The authors use the definition of obesity as BMI >30 kg/m2. However, obesity is defined as BMI >25 kg/m2 for Asian in the Asia-Pacific guidelines.

Reviewer #2: I don't have any objections to the study with respect to any of these questions.

**Results**

-Does the analysis presented match the analysis plan?

-Are the results clearly and completely presented?

-Are the figures (Tables, Images) of sufficient quality for clarity?

Reviewer #1: Yes

Reviewer #2: I think the manuscript is overall okay in this regard. Additional figures, if informative, would be welcome.

**Conclusions**

-Are the conclusions supported by the data presented?

-Are the limitations of analysis clearly described?

-Do the authors discuss how these data can be helpful to advance our understanding of the topic under study?

-Is public health relevance addressed?

Reviewer #1: Yes

Reviewer #2: I am satisfied with the conclusions. In my opinion, this study is a valuable contribution. It has public health relevance, and provides needed information on COVID-19 patients who have mild illness.

**Editorial and Data Presentation Modifications?**

Reviewer #1: (No Response)

Reviewer #2: There were numerous typos in the manuscript, and the writing can and should be substantially improved.

**Summary and General Comments**

Reviewer #1: (No Response)

Reviewer #2: I am overall positive about this work. Despite its limitations, I think that it provides useful information about COVID-19 patients with mild illness, for whom the authors argue that details are lacking. The writing needs to be significantly improved. There were numerous errors, and many parts of the paper are worded awkwardly. I have included some comments and suggestions, but the authors should not take them as an exhaustive list. I recommend going over the entire paper again carefully to improve the writing.

Comments and recommended corrections:

- Page 3, line 50: change "was" to "were"

- Page 3, line 52: change "would" to "can"

- Page 3, line 53: change "and early identify vulnerable patients" to "and identify vulnerable patients in a timely manner"

- Page 3, line 58: change "momentously impact on the health systems of affected countries" to "has had a great impact on the health systems of affected countries"

- Page 3, line 65: change "due to a number" to "due to the fact that a number"

- Page 4, line 77: change "not only more" to "not only the more"

- Page 4, line 78: change "and early identify vulnerable patients" to "and identify vulnerable patients in a timely manner"

- Page 4, line 84: change "was aimed" to "aimed"

- Page 5, line 93: change "were all admitted" to "were admitted"

- Page 5, line 102: change "who" to "who were"

- Page 6, lines 130-131: fix typo

- Page 7, lines 153-154: change "as recovery" to "recoveries"

- Page 7, line 155: change "as death" to "deaths"

- Page 8, line 162: change "defined as patients" to "defined as when patients"

- Page 8, line 167: The authors note that no imputation was made for missing data. Can they describe in detail how in fact they handled missing data?

- Page 8, line 174: should the "and" be "or"?

- Page 8, line 177: Don't start a sentence with "p < 0.05". Find a way to reword this.

- Page 8, lines 183-184: reword "were reached the study outcomes"

- Page 8, lines 190: change "which found" to "which was found"

- Page 16, line 246: I would reword this as "Clinical Outcome and Factories Associated with Pneumonia." Using "predicting" can be problematic.

- Page 23, lines 316-317: change "were recovered and able to discharge" to "recovered and were able to discharge"

- Page 23, line 325: "present with mild illness" is awkward wording

- Page 24, line 328: change "single center may not" to "single center and may not"

- Page 24, line 339: change "In conclusions" to "In conclusion"
---

## [Editor Report · Decision Letter 1]

21 Sep 2020

 Dear Dr. Wiboonchutikul,

We are pleased to inform you that your manuscript 'Clinical course and potential predictive factors for pneumonia of adult patients with Coronavirus Disease 2019 (COVID-19): a retrospective observational analysis of 193 confirmed cases in Thailand' has been provisionally accepted for publication in PLOS Neglected Tropical Diseases.

Best regards,

Johan Van Weyenbergh

Associate Editor

Victor Santana Santos

Deputy Editor

---

## [Editor Report · Acceptance letter]

7 Oct 2020

Dear Dr Wiboonchutikul,

We are delighted to inform you that your manuscript, "Clinical course and potential predictive factors for pneumonia of adult patients with Coronavirus Disease 2019 (COVID-19): a retrospective observational analysis of 193 confirmed cases in Thailand," has been formally accepted for publication in PLOS Neglected Tropical Diseases.

Best regards,

Shaden Kamhawi

co-Editor-in-Chief

Paul Brindley

co-Editor-in-Chief
